# Incidental vocabulary learning with subtitles in a new language: Orthographic markedness and number of exposures

**Mercedes Pérez-Serrano** [1,2]*, **Marta Nogueroles-López** [2,3], **Jon Andoni Duñabeitia** [4,5]

**1** Departamento de Lengua Española y Teoría de la Literatura, Universidad Complutense de Madrid, Madrid, Comunidad de Madrid, Spain, **2** Departamento de Lenguas Aplicadas, Universidad Antonio de Nebrija, Madrid, Comunidad de Madrid, Spain, **3** Departamento de Filología, Comunicación y Documentación, Universidad de Alcalá, Alcalá de Henares, Comunidad de Madrid, Spain, **4** Centro de Ciencia Cognitiva (C3), Facultad de Lenguas y Educación, Universidad Antonio de Nebrija, Madrid, Comunidad de Madrid, Spain, **5** AcqVA Aurora Center, Department of Language and Culture, The Arctic University of Norway, Tromsø, Norway

* mperezserrano@ucm.es

**Data Availability Statement:** All relevant data are within the manuscript.

**Funding:** This study was partially supported by grants RED2018-102615-T and PGC2018-097145-

## Abstract

The present study is set to explore the way the orthographic distributional properties of novel written words and the number of exposures to these words affect their incidental learning in terms of recall and recognition. To that end, two experiments were conducted using videos with captions. These videos included written nonwords (orthographically marked language-specific items) and pseudowords (orthographically unmarked items) as captions paired to the spoken targets, presented either in isolation (Experiment 1) or within sentences (Experiment 2). Our results consistently show that items containing legal letter combinations (i.e., pseudowords) are better recalled and recognized than those with illegal combinations (i.e., nonwords). Further analysis in the recall task indicate that frequency modulates the learning of pseudowords and nonwords in a different way. The learning of pseudowords increases linearly with repetitions, while nonwords are equally learned across frequencies. These differential effects found in the recall task do not show up in the recognition task. Although participants took more time to recognize nonwords in the recognition task, increased exposure to the items similarly modulated reading times and accuracy for nonwords and pseudowords. Additionally, higher accuracy rates were found in Experiment 2, which underscores the beneficial effect of supportive visual information.

## Introduction

The acquisition of new vocabulary is a foundational part of language learning, and the only requirement to learn vocabulary is input [1]. There is general agreement that a significant percentage of vocabulary is acquired incidentally, without the objective of learning, that is through mere exposure to the language. In fact, in one's native language, incidental vocabulary acquisition is regarded as the primary mode of vocabulary acquisition [see 2, 3].

B-I00 from the Spanish Government, and H2019/
HUM-5705 from the Comunidad de Madrid.

**Competing interests:** The authors have declared
that no competing interests exist.

Some factors have been found to have an impact on the incidental acquisition of vocabulary from a foreign language, such as the type of word [4, 5], the morphological predictability of the words [5], the association of the words with pictorial material depicting the concepts [6], and the number of exposures to those new words [7]. In the current study, we explored how some properties of the nonnative words that have been found to change the manner in which they are processed and learned modulate foreign word incidental vocabulary learning. To do so, we focused on orthographic markedness, which refers to effects found in the context of the presence of particular letters or letter combinations in a particular language that are not plausible in another language known to the learner (e.g., the combination of the graphemes C and K does not exist in Spanish, while it does in English). By comparing the incidental learning of marked and unmarked words and how repetition or frequency effects modulate it (i.e., the number of times a given item has been presented during an exposure phase), we aimed at gaining knowledge on how these two factors interact during incidental foreign word learning and their impact in future recall and recognition of the words.

In general, vocabulary learning requires input to happen, be it written, oral or bimodal [1]. Hence, incidental vocabulary also requires a form of exposure to unimodal or multimodal input that will ultimately allow learning a consequence of a gradual process that involves the accumulation of knowledge through the repeated encounters with the words. After the first few encounters, speakers typically learn the word form, but it is not until a higher number of encounters have taken place that the connection between form and meaning stabilizes, since that requires a greater number of repetitions [8, 9].

A great deal of previous studies on incidental vocabulary learning has focused on frequency of appearance in written texts. Preceding research has shown that incidental vocabulary learning through **reading** occurs for both native language learners [e.g. 2, 10] and second language learners [e.g. 11, 12]. It is now well established that readers need many encounters with novel single words that are set in meaningful contexts in order to observe vocabulary knowledge growth as a function of reading [9, 13–15]. However, it should be considered that word knowledge is a multidimensional construct [16], and that the amount of exposure to or encounters with a new item needed to develop a certain type of word knowledge will not be uniform. For example, according to Webb [9], more than ten repetitions may be needed to develop full knowledge of a word, whereas as little as one encounter might suffice to observe gains in receptive lexical knowledge [9, 15].

Recently, Hulme, Barsky and Rodd [17] investigated whether adult readers could learn new meanings incidentally attributed to known words while reading stories and explored the role of the frequency of exposure (namely, the number of encounters; two, four, six, or eight times) to the novel word meaning. Results showed that incidental learning while reading developed linearly with an increase in the number of exposures, following a cumulative incremental trajectory. These findings demonstrate the remarkable success with which adults learn new meanings for known words incidentally while completing a common everyday action like reading.

Likewise, **auditory processing** has also proved to be a good source of incidental vocabulary growth [5, 18–20]. As in the case of the reading studies, research on auditory processing suggests that frequency is a strong predictor of vocabulary learning while listening. Learners can gain knowledge of novel words related to their meaning, grammatical functions and collocations while listening to sentences or speech [21]. Nonetheless, some differences have been also shown between auditory processing and reading in incidental learning. Overall, when compared to learning while reading, learning while listening is less effective and far more encounters are needed. Brown, Waring and Donkaewbua [18] concluded that it is very unlikely for a word to be learnt just by listening to it less than 20 times [see also 19]. Feng and Webb [22]

pointed out that this disadvantage of aural input against written input in incidental vocabulary learning could respond to the faster processing demands required by listening, making it more challenging for learners to notice unknown words in this modality [see also 23].

With this in mind, **audiovisual integration** using bimodal input during incidental word learning could serve to boost the benefits of the two modalities. Studies focusing on incidental learning while participants follow the written words as they listen to the oral text suggest that this is indeed the case [see 7, 13, 18, 24–26]. Bisson, Van Heuven, Conklin, and Tunney [7] investigated the effect of repeated exposures to multimodal stimuli on the incidental acquisition of new vocabulary from the foreign language by manipulating the number of appearances of the new words (2, 4, 6, and 8 exposures). Following an incidental learning phase, participants completed an explicit learning task in which they learned the mappings between translation equivalents in their native and foreign language. Critically, some of the items had been previously presented in the incidental learning phase, while others were totally new. The results showed that participants performed better on the words they had previously been exposed to, and that this incidental learning effect took place even for items that were only presented twice.

The importance of exposure to bimodal input for foreign language incidental learning should be further considered, given that it opens doors to ecologically valid learning settings that could boost vocabulary acquisition. Watching television or videos is beneficial for foreign language learners' vocabulary growth [e.g. 27, 28], and research has shown that vocabulary learning in these contexts is also governed by frequency effects [29]. On-screen text aids such as native or foreign language subtitles or captions could be effective ways to turn video materials into effective didactic tools for foreign language incidental word learning, given the evidence showing their effectiveness in vocabulary learning [27, 29, 30]. Hence, it is worth exploring the mediating role of the amount of exposures (i.e., the frequency effect) to a new item in audiovisual settings with bimodal presentation to understand how incidental vocabulary learning occurs in naturalistic learning contexts. The current study was set to provide evidence in this regard.

Over and above external manipulations involving the amount of exposure to a series of items, additional factors that are intrinsic to these items are also expected to modulate incidental learning effects. One of these factors that has been extensively studied in recent years in the domain of multilingual language processing is the orthotactic pattern of a written item. The orthotactics of a word corresponds to the distributional properties of the orthographic representations of that word (e.g., letters or bigrams) within the language it belongs to, and also across the languages known to a bilingual person. The orthotactic properties of a word may hint at the language to which it belongs. For instance, in the case of an English-Spanish bilingual, words with a WH-onset would point to English lexemes.

Preceding studies have demonstrated that bilinguals benefit from the presence of orthographic cues (namely, orthographic markedness) to detect the language of a given written word [31–35]. When bilingual readers are required to identify as quickly as possible the language of a series of stimuli they are presented with, response latencies are faster and accuracy rates are higher for orthographically marked than unmarked words [31, 34]. These findings, together with recent evidence from neuroimaging data [see 35] demonstrates that language-specific orthography guides single word identification, and thus suggests that access to the lexicon might be guided by the extraction of language-specific orthotactic combinatorial rules [32].

Focusing specifically on the distinction between incidental and intentional vocabulary learning, Bordag, Kirschenbaum, Rogahn and Tschirner [36] examined the role of orthotactics (understood as the relative frequencies of the letters and their sequences within words) in the

early stages of vocabulary acquisition by adult native German speakers and advanced learners of German. They explored in which way low versus high orthotactic probabilities affected the acquisition of the meaning of new words in a native and foreign language and whether the effect differed between intentional and incidental acquisition modes. On the one hand, the results showed that low orthotactic probabilities (namely, high orthographic markedness levels) contributed positively to incidental acquisition of novel word meanings in the first language, but no such effect was found in the foreign language. That is, the meaning of words from the native language was incidentally acquired better when these words were orthographically marked. On the other hand, high orthographic probabilities (namely, low orthographic markedness levels) positively affected intentional learning in a foreign language, but not in the native language.

The current study was set in order to shed further light on the way the orthographic distributional properties of novel written words impact their incidental learning in a setting that could resemble to those used in foreign language learning contexts. Besides, the manner in which orthographic markedness and number of encounters or frequency interact during incidental word learning and how they modulate the ability to recognize and recall these words was also explored. To this end, and following the line of research using subtitles and captions to study incidental language learning, two experiments were conducted using videos with captions. These videos included written nonwords (orthographically marked language-specific items) and pseudowords (orthographically unmarked items) as captions paired to the spoken targets, presented either in isolation (Experiment 1) or within sentences (Experiment 2). Using nonwords and pseudowords enabled us to control for the prior lexical knowledge of the participants. We were especially interested in exploring to what extent novel words that are orthotactically illegal in the native language would benefit from a higher number of encounters during incidental learning settings.

## Experiment 1: Captions of isolated novel words

In Experiment 1, participants were visually presented with two videos containing images that were unrelated to the synchronously presented audio tracks containing information about some animals and some fruits. Each of the target animal and fruit name presented auditorily was paired with a written element that was displayed on the screen, consisting either of a pseudoword or a nonword. Critically, some of the elements were presented more often than others (namely, a frequency manipulation). After the passive perception of the videos and soundtracks, participants completed a recall and a recognition test.

## Materials and methods

### Participants

Forty Spanish students (33 women; mean age = 19.6 years) majoring in Modern Languages and Psychology at Nebrija University took part in the experiment. All the students had Spanish as their native language and participated voluntarily in exchange for monetary compensation. Prior to the experimental session, all participants gave their informed consent in accordance with guidelines approved by the Ethics and Research Committees of the Nebrija University.

### Design and stimuli

We used a repeated-measures design for this experiment with Type of Item (nonword | pseudoword) and Number of Exposures (1 | 4 | 8) as within-subject factors. The stimuli used were two videos created by the researchers with successive images of landscapes that did not include

any piece of information that could be related in any way to the content of the soundtrack. The soundtracks of each of the two videos consisted of 28 sentences in Spanish containing a total of 12 targets. One of the soundtracks presented information about 6 animals (cat, mouse, rabbit, bird, fish and turtle), and the other soundtrack referred to 6 fruits (mango, pineapple, melon, pear, grape, strawberry). By creating the material *ex profeso*, we ensured having full control over the factors that could influence the results, such as the precise number of exposures to the targets and their characteristics.

Each target word was paired with a corresponding nonword (orthotactically marked) or pseudoword (orthotactically unmarked) as shown in Table 1.

Each soundtrack consisted of 28 sentences. 26 sentences contained a single mention to one of the Spanish targets, and one opening and one closing sentences were added too. The critical 26 sentences comprised objective information on the fruits and animals. The two texts that were used to create the soundtracks were highly similar in length (fruits: 488 words; animals: 479 words) and had a very similar structure. The texts were transformed into speech using an online software and the resulting audio clips were included in the videos using video editing software. Three different versions of each final video clip were created in order to ensure that the place of appearance of the targets did not affect the results. To do so, the order of appearance of the sentences in the audio files was randomized. The resulting 6 video clips ranged from 3:33 to 3:41 minutes in length. Each participant was randomly assigned one version of each video. The pseudowords and nonwords appeared on the lower central part of the screen during the videos when the corresponding target was produced in the audios. They were written in uppercase and displayed for 2 seconds. As shown in Table 1, the items were split into three subsets such that words were presented either once, four times, or eight times in each video clip. For every one of the levels of the Number of Exposure factor, each video clip included one specific pairing between a Spanish word and a nonword (orthotactically marked), and another pairing of a different Spanish word and a pseudoword (orthotactically unmarked). The pseudowords were created in accordance with the orthotactic rules of Spanish, and contained legal bigrams in this language (e.g., *mastu*). The nonwords did not follow the Spanish orthotactic rules and each contained an illegal bigram in Spanish (e.g., the bigram ZF in *rezfa*). All the nonwords and pseudowords were pronounceable disyllabic 5-letter strings with the same orthographic structure (namely, CVCCV).

**Table 1. Target words paired with a nonword (NW) or pseudoword (PW) item.**

| Target Word | String | Number of Exposures |
|---|---|---|
| *FRUITS* | | |
| Strawberry | vatre (PW) | 8 |
| Grape | newza (NW) | 8 |
| Melon | dinse (PW) | 4 |
| Mango | gojri (NW) | 4 |
| Pineapple | zuspo (PW) | 1 |
| Pear | begxu (NW) | 1 |
| *ANIMALS* | | |
| Rabbit | mastu (PW) | 8 |
| Mouse | rezfa (NW) | 8 |
| Bird | linga (PW) | 4 |
| Fish | rojle (NW) | 4 |
| Cat | tumpi (PW) | 1 |
| Turtle | camgo (NW) | 1 |

## Procedure

The experiment was created using Gorilla [37]. All participants were tested in individual PCs with headsets. After signing the consent forms, the participants completed a short sociodemographic questionnaire, and then they watched the two corresponding video clips consecutively. The participants were naive to the phases of the experiment and the experimenters did not mention the subtitles in the videos or the tasks that would come after. The only instruction that the experimenters gave to the participants was to pay attention to what was played on the screen. This provided us with an incidental learning situation in which participants could associate the aural input (namely, the Spanish words for the fruits and animals) with the on-screen text (namely, the pseudowords and the nonwords). This allowed us to operationalize the construct of awareness non-concurrently at the reconstruction stage [38].

Once the two video clips had ended, participants completed a simple working memory test (a timed N-back task in which they had to indicate via button presses whether or not a given letter had already been presented two trials before). This test was aimed at diverting the attention from the content of the video clips and was used as a filler task. At the end of the N-back task, participants were asked to complete the two critical tests that were devised to measure recall and recognition of the correspondences between the displayed new items (pseudowords and nonwords) and the played Spanish words (animal and fruit names). First, the recall test was completed. In this task, participants were presented with the picture and the Spanish name of each of the fruits and animals used in the videos, and they were required to write the name they think they saw during the video accompanying each of them. Each trial started with the centered presentation of a fixation cross for 1000ms, immediately followed by the concurrent presentation of a picture and the lowercase Spanish name of one of the items that were mentioned in the video clips. Participants had to type the string they remembered, and they were explicitly told that they could not advance without typing anything. Once they had pressed the enter button, they had to complete a trial from a recognition task, in which another fixation cross was displayed for 500ms, followed by the same picture and Spanish word in the same location, this time accompanied by two alternatives to select from. The two displayed alternatives were lowercase strings corresponding to the correct string (namely, the one matching the target) and a competitor taken from the same video that had been presented in the exposure phase the same number of times. For instance, when the picture of a mouse was displayed together with the Spanish word *ratón*, the correct target *rezfa* was displayed with the mismatching alternative *mastu*, since the latter was presented the same number of times in the video (8 times, in this example). Participants completed the 2-alternative forced choice (2AFC) task without time pressure, and accuracy was prioritized and no time limit was set for the responses. The whole experiment lasted for about 20 minutes.

## Results and discussion

An analysis of the data from the recall and the recognition tests was carried out for the group of 40 participants with Type of Item (nonword | pseudoword) and Number of Exposures (1 | 4 | 8) as within-subject factors in a series of ANOVAs. Descriptive statistics split per condition are reported in Table 2.

### Recall test

In a first analysis, we explored the success or failure in the recall task by considering exclusively the coincidence between the typed strings and the expected ones. As shown in Table 2, the exact recall was relatively low in general, with a mean error rate of 87.9% (SD = 16.1, range: 41.7–100). The ANOVA showed a significant main effect of Type of Item (F(1,39) = 8.86, p =

**Table 2. Descriptive statistics for each of the dependent variables in each of the conditions tested in the recall and recognition tests.**

| | Pseudowords | | | Nonwords | | |
|---|---|---|---|---|---|---|
| | 1 | 4 | 8 | 1 | 4 | 8 |
| Percentage of errors in recall | 97.5 (11.0) | 85.0 (28.2) | 68.8 (37.0) | 95.0 (18.9) | 90.0 (20.3) | 91.3 (22.3) |
| Levenshtein Distance (number of edits) | 4.59 (0.74) | 3.69 (1.61) | 2.75 (1.94) | 4.36 (1.03) | 3.67 (1.44) | 3.42 (1.50) |
| Percentage of errors in recognition | 47.5 (33.9) | 26.3 (33.9) | 22.5 (31.9) | 37.5 (37.1) | 32.5 (33.1) | 18.8 (29.3) |
| Reaction Times (milliseconds) | 2371 (947) | 2061 (1075) | 1794 (719) | 2313 (1022) | 2556 (1006) | 2170 (854) |

Standard deviations are presented in parenthesis.

.005), demonstrating that pseudowords were recalled with higher accuracy levels than nonwords. The main effect of the Number of Exposures was also significant (F(1,39) = 13.28, p < .001), showing that the recall increased as a function of repetitions. More importantly, the interaction between the two factors was significant (F(1,78) = 9.04, p < .001). Follow-up post hoc tests (Holm-corrected pairwise comparisons) showed that the recall for the pseudowords increased with repetitions (1 vs. 4: t(156) = 2.86, p = .048; 4 vs. 8: t(156) = 3.72, p = .003). However, this was not the case for nonword items, since no differences were found as a function of increased exposure (all ts<1.2 and ps>.99; see Fig 1).

Admittedly, precise recall in an incidental vocabulary learning task such as the present one may not be expected. With this in mind, we carried out an additional analysis on the recall data based on the Levenshtein distance (LD) between the participants' responses and the actual targets. LD is an index of the number of edits (replacements, insertions or additions) needed to get from one text string to another, and thus allows for an estimation of the degree of similarity between the typed response and the expected one, with larger values representing greater deviance. As shown in Table 2, the mean deviance was of 3.75 edits (SD = 1.03). An ANOVA with the same factors and levels was carried out and results revealed a significant Number of Exposures effect (F(2,78) = 29.59, p < .001) and no effect of Type of Item (F<1 and p>.35). Critically, the interaction between the two factors was significant (F(2,78) = 4.88, p = .010). Holm-corrected pairwise comparisons showed that the number of edits needed to convert the typed responses into the expected ones decreased as a function of exposure, but that this decrease was significantly more marked for pseudowords than for nonwords (pseudowords:

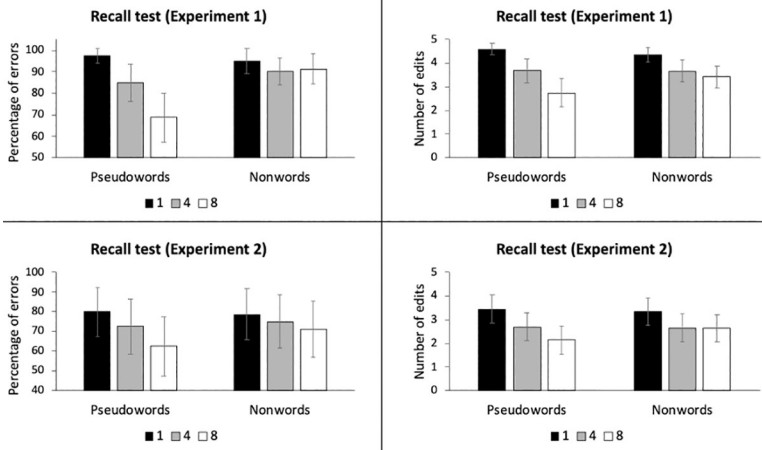

**Fig 1. Results in the recall tasks of Experiment 1 (upper panels) and Experiment 2 (lower panels): Percentage of errors (left panels) and number of edits (right panels).** Error bars correspond to the 95% confidence intervals.

1vs. 4: t(151) = 3.83, p = .002; 4 vs. 8: t(151) = 3.98, p = .001; nonwords: 1 vs. 4: t(151) = 2.92, p = .028; 4 vs. 8: t(151) = 1.06, p>.99; see Table 2 and Fig 1).

### Recognition test

The results of the accuracy rates in the recognition test were analyzed following the same design. Results showed that participants recognized the correct responses with a good level of accuracy, since the mean percentage of errors was fairly low (Mean = 30.8%, SD = 19.7; see Table 2). The ANOVA showed a significant main effect of Number of Exposures, so that recognition accuracy increased as a function of repetitions (F(2,78) = 8.27, p < .001). The main effect of Type of Item and the interaction between the two factors were not significant (Fs<2 and ps>.16).

A parallel analysis was carried out on the latency data to shed further light on the effects, even if participants were not told to provide speeded responses. Reaction times associated with erroneous responses were not included in the analysis, and latencies above 5000ms or beyond 300ms were also discarded (representing only 4.82% of the data). The ANOVA showed a main effect of Type of Item (F(1,20) = 8.17, p = .010), demonstrating that pseudoword items were recognized faster than nonword items (see Table 2 for descriptives). The effect of Number of Exposures was also significant (F(2,40) = 4.13, p = .023), showing that reaction times varied as an inverse function of exposure. As in the case of the accuracy data, the interaction between the two factors was not significant (F<1 and p>.40).

In terms of recall, our results confirm that orthotactically unmarked novel words (namely, the pseudowords) are better retrieved and benefit more from an increased exposure than orthotactically marked words (namely, the nonwords). Across measures, our results demonstrate a main frequency effect showing an increase in recall performance as a function of repetitions (i.e., strings encountered more times showed better recall patterns). More importantly, this frequency effect was significantly more marked for pseudowords than for nonwords (with the latter type of strings showing negligible frequency effects in direct recall). The analysis of the recognition task showed a different pattern, suggesting that accuracy improved with repetitions, and that reaction times also decreased as a function of repetitions. A general recognition latency benefit was found for legal novel words over illegal strings, but no clear signs of an interaction was found either in the accuracy or in the response times.

One issue that should be kept in mind is that the videos that participants viewed did not contain any visual information directly related with the auditorily presented information. This was done in order to artificially create an incidental learning scenario, but this admittedly created an unnatural set-up that notably deviates from real life settings (e.g., watching a movie with subtitles). Besides, the presentation of the nonwords and pseudowords in isolation could have made their processing more salient, given that these were the only written elements displayed. And if this were the case, the fully incidental nature of the learning process could be questioned. For these reasons, and in an attempt to replicate the findings and extend them to more realistic scenarios, a new experiment was conducted.

## Experiment 2: Captions with novel words embedded

In Experiment 2, participants were presented with two videos including the same verbal content as in Experiment 1, but this time accompanied by supporting related visual information depicting the animals and fruits mentioned. Besides, the captions of individual novel words used in Experiment 1 were replaced by captions of the whole sentences in which the critical words had been changed for the novel words (either nonwords or pseudowords). By making

these changes, we intended to create a more realistic set-up and an incidental learning scenario.

## Materials and methods

### Participants

This second experiment was set up with the participation of 40 graduate and postgraduate students at Nebrija University (31 women; mean age = 30.4 years). As in Experiment 1, all participants gave their informed consent in accordance with guidelines approved by the Ethics and Research Committees of the Nebrija University.

### Design and stimuli

The same design used in Experiment 1 was also used in Experiment 2, with Type of Item (nonword | pseudoword) and Number of Exposures (1 | 4 | 8) as within-subject factors. The exact same manipulations done in Experiment 1 were also used in Experiment 2 (namely, presenting some items once, four times, or eight times in each video clip). To this end, the same verbal content was used for the soundtracks, and the critical novel words were those presented in Table 1.

The stimuli used were two videos created by the researchers, with the corresponding soundtracks. The soundtrack of each of the two videos consisted of 28 sentences in Spanish, 26 of which provided information regarding the critical words. The remaining 2 sentences were included for opening and closing purposes. Hence, the critical sentences of each of the soundtracks reported statements about the fruits and the animals. Differently from Experiment 1, the visual content of the videos used in Experiment 2 depicted the information contained in the soundtracks, consisting of successive images of fruits and animals. One video presented information about 6 animals (cat, mouse, rabbit, bird, fish and turtle), and the other one referred to 6 fruits (mango, pineapple, melon, pear, grape, strawberry). As in Experiment 1, each target word was paired with either a nonword (orthotactically marked) or pseudoword (orthotactically unmarked; see Table 1). Two different versions of each video clip were created by randomizing the order of appearance of the sentences and the images, and participants were randomly assigned to them. Similarly, the order of presentation of the videos (fruits and animals) was counterbalanced across participants.

The subtitles accompanying each image and spoken sentence displayed the entire utterance with the novel word (nonword or pseudoword) embedded, substituting the real word, as a way to facilitate the integration of its meaning (e.g., *La vida en cautividad de un REZFA puede llegar a los dos años*, [*The life in captivity of a REZFA can reach two years*], where REZFA substitutes the word RATÓN [mouse] that is presented auditorily paired with the image of a mouse). The images, the audio tracks and the on-screen text were displayed synchronically, so that the subtitles, and consequently the novel words, were displayed for the duration of the corresponding utterance.

### Procedure

The procedure was similar to that used in Experiment 1 except for two changes that were implemented. First, the filler N-back task used in Experiment 1 was replaced by a non-linguistic version of the task in which participants were presented with a sequence of colored figures and they had to indicate whether or not a given figure had already been presented two trials before. This change was done in order to avoid any potential effect of a linguistic filler task on the recall and recognition tests. And second, participants first completed the recall test and

once they had finished it, they started the recognition task. This was done in order to avoid any specific testing effect or interaction between task demands.

## Results and discussion

An analysis of the data from the recall and the recognition tests was carried out following the same procedure as in Experiment 1, with Type of Item (nonword | pseudoword) and Number of Exposures (1 | 4 | 8) as within-subject factors in a series of ANOVAs. Descriptive statistics are reported in Table 3.

### Recall test

As shown in Table 3, the exact recall was moderate in general, with a mean error rate of 73.3% (SD = 44.3). The ANOVA showed a significant main effect of Number of Exposures ($F_{(2,78)}$ = 5.20, p = .008), suggesting that recall increased as a function of repetitions. The main effect of Type of Item and the interaction between the two factors did not result significant (F<2, p>.19).

The analysis of the number of edits needed to convert each answer given by the participants into the correct target string (the Levenshtein distance (LD) measure) showed that the mean deviance was of 2.82 edits (SD = 1.88). An ANOVA revealed a significant Number of Exposures effect ($F_{(2,78)}$ = 16.92, p < .001) and no effect of Type of Item (F<1.5 and p>.25). The interaction between the two factors was significant ($F_{(2,78)}$ = 3.33, p = .041). Holm-corrected pairwise comparisons showed that the number of edits needed to convert the typed responses into the expected ones decreased as a function of exposure, but that this decrease was significantly more marked for pseudowords than for nonwords (pseudowords: 1vs. 4: $t_{(138)}$ = 3.49, p = .007; 4 vs. 8: $t_{(138)}$ = 2.51, p = .078; nonwords: 1 vs. 4: $t_{(138)}$ = 3.14, p = .018; 4 vs. 8: $t_{(138)}$ = 0.17, p>.99; see Table 3 and Fig 1).

### Recognition test

The mean percentage of errors in the recognition task was relatively low, with a mean of 21.9% errors (SD = 30; see Table 3), suggesting that the level of accuracy was good in terms of recognition. The ANOVA showed a significant main effect of Number of Exposures, so that recognition accuracy increased as a function of repetitions ($F_{(2,78)}$ = 7.08, p < .001). The main effect of Type of Item and the interaction between the two factors were not significant (Fs<1 and ps>.55).

As for reaction times, results were consistent with those obtained in Experiment 1, since the ANOVA also showed a main effect of Type of Item ($F_{(1,30)}$ = 5.87, p = .022), demonstrating that pseudoword items were recognized faster than nonword items (see Table 3). The effect of

**Table 3. Experiment 2.** Descriptive statistics for each of the dependent variables in each of the conditions tested in the recall and recognition tests.

|  | Pseudowords | | | Nonwords | | |
|---|---|---|---|---|---|---|
|  | 1 | 4 | 8 | 1 | 4 | 8 |
| **Percentage of errors in recall** | 80.0 (40.3) | 72.5 (44.9) | 62.5 (48.7) | 78.8 (41.2) | 75.0 (43.6) | 71.3 (45.5) |
| **Levenshtein Distance (number of edits)** | 3.45 (1.95) | 2.69 (1.92) | 2.14 (1.87) | 3.35 (1.81) | 2.66 (1.87) | 2.63 (1.86) |
| **Percentage of errors in recognition** | 28.8 (27.5) | 20.0 (33.6) | 18.8 (29.3) | 31.3 (37.0) | 15.0 (25.8) | 17.5 (26.7) |
| **Reaction Times (milliseconds)** | 3185 (1951) | 2722 (1349) | 2498 (1354) | 3040 (1529) | 2882 (1776) | 2963 (1580) |

Standard deviations are presented in parenthesis.

Number of Exposures and the interaction between the two factors were not significant (F<1 and p>.42).

The data obtained in Experiment 2 confirm the observation from Experiment 1 suggesting that the more times a novel word is encountered, the better this item is later recalled. As in Experiment 1, the current experiment demonstrated that this frequency effect is especially salient for new words that align with the orthotactics of the known language (namely, pseudo-words). The results in terms of recognition were also consistent with those obtained in Experiment 1, showing that the percentages of errors decreased as a function of the number of repetitions, and that the reaction times were significantly faster for pseudowords than for nonwords.

## General discussion

Our results support previous research in second language acquisition which links frequency of appearance and incidental vocabulary learning. In the current study, we found an effect of the frequency of the incidental exposure on the new words in their later recall and recognition. While the incidental nature of the two experiments presented here was clearly different (with Experiment 2 representing a more conventional incidental learning scenario than Experiment 1), the results regarding the frequency effects were highly similar, showing a better recall pattern and a more accurate recognition pattern of strings that have been perceived more often. Preceding studies have demonstrated that frequency fosters vocabulary learning in combination with second language subtitles and captions [29], and a positive correlation between the number of appearances of new word forms and vocabulary learning while watching TV without subtitles in a foreign language has been reported [28]. However, and to the best of our knowledge, this study is the first one addressing the role of frequency of incidental vocabulary learning while processing reversed subtitled clips, demonstrating the usefulness of controlling for the number of appearances of new lexical items in incidental learning scenarios as the ones presented here. These results align with preceding studies on incidental vocabulary learning showing that the more times a word is encountered, the higher the chances of it being learnt are across modalities [in reading: 15, 17; listening: 19; listening while watching pictures: 6; and listening while reading: 18, 26].

Besides, this is the first study exploring the interactions between the number of repetitions of a given element and its orthotactic characteristics, and how these factors drive incidental learning. In line with other studies like those by Ellis and Beaton [39] and Borragán de Bruin, Havas, de Diego-Balaguer, Vulchanova, Vulchanov, et al. [40], words that are orthotactically similar to the participants' L1 are better recalled, while those that are orthotactically different present recall difficulties. Results from Experiments 1 and 2 consistently show that the recall of items containing letter combinations that match the orthotactic pattern of the native language increase as a function of repeated exposures much more than the recall of orthographically marked novel words. Hence, our results demonstrate that the recall of incidentally learned novel words that follow the native language's orthotactics increases linearly with repetitions, while the recall of newly learned nonwords is not affected by frequency to the same extent. Items that include language-specific orthographic cues (namely, nonwords) do not show marked learning gains in terms of recall across all frequencies. These results suggest that the orthotactic structure of the novel words represents a critical factor for their learning, especially after the initial exposure to them. New words that are orthotactically similar to those from the native language benefit more from subsequent repetitions in a short span, leading to strong and reliable frequency effects. Words that deviate from the orthotactic structure of the known language, on the other hand, might need more explicit attention.

Hulme, Barsky and Rodd [17] obtained that a good portion (around 38%) of readers could correctly recall a new meaning for a known word after just two exposures to it in a written story when they were tested immediately after the training phase. They noticed that recall accuracy improved as the number of exposures increased, and that the first two encounters of a word (the first two exposures to it) are the ones determining their learning to a larger extent. Based on those results, Hulme, Barsky and Rodd claimed that first encounters are decisive, especially for the acquisition of homonyms, and they interpreted these results as being the consequence of the longer time spent in reading and processing words during the initial exposures to them [see 41, 42]. Nonetheless, these studies did not measure active recall (namely, explicitly providing the newly learned word form), which is the most stressful and complex aspect of word knowledge [e.g., 43, 44]. In the context tested in the current study, we demonstrated that effective and accurate recall only took place for words that respected the orthographic properties of the native language, and only after some repetitions were introduced. These results also align with those reported by Vander Beken and Brysbaert [45] who tested free recall from short expository texts studied in a native and in a second language and found that the latter case impeded free recall test as compared to the former. Likewise, in the present study participants showed increased difficulty in recalling nonwords with bigrams which are not common in their native language (namely, items that include bigram combinations that are specific to the new language) as compared to pseudowords. Such differences in recall between pseudowords and nonwords might be due, as Vander Beken and Brysbaert [45] suggested, to the encoding complexity associated to the latter type of strings and to the difficulty that free recall tasks involve for learners. As demonstrated by Vander Beken and Brysbaert, learners of a new language typically show difficulties when required to write lexical items in a foreign language, possibly as a consequence of the increased stress involved in recall tasks that are in a language that differs from the native one. In the case of orthographically marked new items that deviate from the native language's orthotactic standards, the inherent difficulty of these items together with task-dependent factors could have been responsible for the observed interactions and differences between conditions.

The effects found in the recognition task were also illustrative and they speak for the importance of strategically distributing the new lexical forms from a new language and their repetitions across the texts that learners receive. A clear-cut frequency effect was found in the recognition tasks, showing that the more times a new word is incidentally met, the more accurately that this word will be later recognized. There is no current agreement on the exact number of exposures required to detect a significant incidental learning effect of new words. Pellicer Sánchez and Schmitt [15] found that one encounter might be enough for ulterior meaning recognition during reading. Webb, Newton, Chang [25] found significant incidental learning with five exposures in the context of reading-while-listening. van Zeeland and Schmitt's [19] suggested that vocabulary learning through oral input only needed more than 15 exposures. Bisson, Van Heuven, Conklin and Tunney [46] found learning effects in a translation recognition task with only two exposures to multimodal stimuli. However, when targets are presented in sentences, as in our study, it is likely that more repetitions are needed for effective vocabulary knowledge to be acquired [e.g., 47]. Our results suggest that the recognition accuracy for newly learned items increases following a relatively linear trend of around 10% every 4 additional exposures (see Tables 2 and 3). These results consistently demonstrated that participants recognized pseudowords faster than nonwords, pointing to a pseudoword recognition superiority effect. These results are in line with those reported by Casaponsa and Duñabeitia [32], who found that letters embedded in strings that respected the orthotactics of the native language (i.e., unmarked language-unspecific strings) were recognized faster than letters embedded in strings containing bigrams that were implausible in the native language.

In this respect, we found differences between the recognition of pseudowords and of nonwords in terms of latency, but not in terms of accuracy (see [32] for a similar pattern).

Unlike the results from the recall tasks, repeated exposures yielded an advantage in recognition for both pseudowords and nonwords. This difference in results between recall and recognition tasks concerning the interaction between frequency and orthotactics might be partially explained by the testing effect, which suggests that tests not only assess learning but also enhance it [48]. It should be kept in mind that the alternative response options presented in the recognition tasks provided learners with additional practice in retrieval processes [45].

Although results obtained in both experiments were highly consistent, we found higher accuracy rates in Experiment 2. Participants in this second experiment were presented with a more ecologically valid incidental learning scenario mimicking naturalistic settings (e.g., watching TV in the L1 with subtitles in an L2 and with visual information matching aural script). The acknowledged link between imagery and vocabulary learning is supported by theories of multimedia learning [49], which state that learning improves when learners can access supportive visual and aural information. Hence, the three types of consistent information channels (namely, the auditory discourse, the written captions and the visual images) could have boosted incidental learning in Experiment 2 as compared with Experiment 1, in which only two sources of supportive information were provided. These findings underscore the beneficial role of imagery in incidental vocabulary learning, as also demonstrated by Bisson, van Heuven, Conklin, and Tunney [6] and Rodgers [50].

Most of the limitations of Experiment 1 were amended in Experiment 2, but some additional issues and questions remain open for future studies. First, the number of items per condition was certainly limited, as it is the case in other preceding studies of incidental word learning in the native and foreign language (see [51]). And second, all targets in the present study were semantically classified into two groups (animals and fruits). In these respect, future research should address the effects of semantic clustering and repetition on incidental vocabulary learning, using different materials and additional items.

In a nutshell, the present study was aimed to investigate the effects of orthographic markedness and of the number of encounters with a word on its incidental learning. The effects of the two factors studied are consistent within the two experiments, although higher learning gains were found for participants involved in more realistic learning situations (Experiment 2). Results showed that the recall of pseudowords was more accurate than that of nonwords and that it increased with repetitions. Repeated nonword items showed a reduced recall advantage, and increased exposure to them did not yield the same recall benefits found for pseudowords. Pseudoword items were also recognized faster than nonwords, and the recognition accuracy of both types increased as a function of the number of exposures. Repetition effects were similar for both pseudowords and nonwords. In summary, these results highlight the role of repetitions for efficient incidental learning. Besides, they also underline the impact of the orthographic similarity between the words from a new language and their corresponding translation equivalents in the known language in their incidental learning through repeated exposures.

## Supporting information

**S1 Appendix. Aural texts.**
(DOCX)

## Author Contributions

**Conceptualization:** Mercedes Pérez-Serrano, Marta Nogueroles-López, Jon Andoni Duñabeitia.

**Data curation:** Mercedes Pérez-Serrano, Marta Nogueroles-López.

**Formal analysis:** Jon Andoni Duñabeitia.

**Funding acquisition:** Jon Andoni Duñabeitia.

**Investigation:** Jon Andoni Duñabeitia.

**Methodology:** Mercedes Pérez-Serrano, Marta Nogueroles-López.

**Resources:** Mercedes Pérez-Serrano.

**Software:** Jon Andoni Duñabeitia.

**Supervision:** Jon Andoni Duñabeitia.

**Writing – original draft:** Mercedes Pérez-Serrano, Marta Nogueroles-López.

**Writing – review & editing:** Mercedes Pérez-Serrano, Marta Nogueroles-López, Jon Andoni Duñabeitia.

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
