## [Decision Letter · Decision Letter 0]

28 Jun 2020

PONE-D-20-13468

Incidental vocabulary learning with subtitles in a new language: Orthographic markedness and number of exposures

PLOS ONE

Dear Dr. Pérez Serrano,

Thank you for submitting your manuscript to PLOS ONE. Two experts have now reviewed your manuscript. As you will see from their comments, both reviewers found merit in your research but also raised a number of questions that you will need to carefully take into consideration before the manuscript is ready for publication. The authors should pay particular attention to the relationship between their design and incidental learning, an issue which might require collecting new data. In sum, i am happy to invite you to submit a revised version of the manuscript that addresses the points raised during the review process.

We look forward to receiving your revised manuscript.

Kind regards,

José A Hinojosa, Ph.D.

Academic Editor

PLOS ONE

Journal Requirements:

'This study was partially supported by grants RED2018-102615-T and

PGC2018-097145-B-I00 from the Spanish Government, and H2019/HUM-5705 from the

Comunidad de Madrid.'

 'The funders had no role in study design, data collection and analysis, decision to publish, or preparation of the manuscript.'

Reviewers' comments:

Reviewer's Responses to Questions

**Comments to the Author**

1. Is the manuscript technically sound, and do the data support the conclusions?

Reviewer #1: Partly

Reviewer #2: Partly

2. Has the statistical analysis been performed appropriately and rigorously? 

Reviewer #1: No

Reviewer #2: Yes

3. Have the authors made all data underlying the findings in their manuscript fully available?

Reviewer #1: Yes

Reviewer #2: Yes

4. Is the manuscript presented in an intelligible fashion and written in standard English?

Reviewer #1: Yes

Reviewer #2: Yes

5. Review Comments to the Author

Reviewer #1: Comments on manuscript Incidental vocabulary learning with subtitles in a new language: Orthographic markedness and number of exposures

I like the paper and the idea is interesting, but the experiment suffers from several problems that need to be improved before publication. The easy path would be running a second better controlled experiment.

The procedure does not correspond to an incidental learning situation. Usually, exposures to linguistic stimuli are made through a single modality. In the case subtitled films, this is done by image, sound and text. In this case images (landscapes) were used, but they did not contain references to linguistic material. What then was the function of the images? It could have been done with text and audio only.

The authors showed the target words by text, instead of the entire utterance that would be the logical choice. In this way participants' attention is now drawn to these words, so that learning is no longer incidental.

The number of exposures and the spelling characteristics (pseudoword vs. nonword) are not manipulated through the items. There are very few items per condition. Only F1 is given.

The tasks used for the assessment only measure recall and recognition but not semantic and orthographic integration.

There are no examples of the materials: Do the sentences form a coherent text or are they unrelated sentences?

Reviewer #2: This is a study simulating foreign vocabulary learning. The authors focus on the orthographic markedness of the new words and the number of exposures. The results reveal that both factors have a role on learning. The study

uses a novel approach, consisting of the conjoint exposure to written and auditory information during learning, and may have practical implications for the field of second language acquisition. Thus, it has potential interest for the readers of PlosOne. I have several concerns, however, that preclude the publication of the manuscript in its present form. I list them in chronological order:

Ecological experimental paradigm:

The authors state that their paradigm mimicks incidental vocabulary learning in naturalistic learning contexts. Concretely, they compare the paradigm used here with watching television or videos with subtitles. This similarity is evidenced by the title of the paper “Incidental vocabulary learning with subtitles in a new language”.

I do not agree, however, with this comparison. It seems to me that this procedure does not mimic ecological incidental vocabulary learning. In my opinion, being exposed to an audiotape that has not any relationship with a series of images (landscapes) that are displayed on the screen, where a single word in an unknown language is presented, is not very ecological. This is rather an artificial situation, different from watching films with subtitles, where the auditory and the visual messages are integrated. Therefore, the autors should temper their claims about the ecological nature of the paradigm. I wonder why the authors didn’t use a more realistic situation, in which the content of the soundtrack is integrated with the content of the visual scenes. I would like to know also why they didn’t present the written novel word (pseudoword or nonword) embedded in the Spanish sentence, as a way to facilitate the integration of its meaning.

Participants:

More information about the linguistic profile of the participants is needed. Considering that they are majoring in modern languages, they probably know several languages other than Spanish. Furthermore, they may have a high proficiency level in some of them. This information is relevant for two reasons. On the one hand, it has been demonstrated that bilinguals have a greater facility to acquire vocabulary in an unknown language than monolinguals (Kaushanskaya and Marian, 2009). This might explain the good performance observed in the recognition task (at least in the 4-8 exposures conditions). On the other hand, it is relevant to know if the letter combinations that are not allowed in Spanish (illegal bigrams) are legal combinations in any other of the languages spoken by the participants. If that is the case, this might have reduced the difference between pseudowords and nonwords for some participants.

Materials/Procedure:

-The number of items in each condition is very low. This is a 2x3 design, including a total amount of 12 words. This means that there are only 2 items per condition. I am aware that participants in this type of experiments cannot learn many new words, but this does not justify such a small number of items. In my opinion, this is the main limitation of the study. The authors might have used a more extensive training procedure, as other authors have done. Alternatively, they might have manipulated the frequency of exposure in a between-participants design.

-The results obtained here with such a small number of items may be restricted to this particular set of stimuli. The authors have to carry out a by-items analysis to examine if the effects can be generalized over items. This analysis has to be included in the paper, together with the by-participants analysis.

-All the participants were presented with the same 4 novel words in the 1, 4 and 8 exposures conditions. I wonder why the authors didn’t counterbalance the set of words included in these three conditions across participants. This would have been the most suitable approach to avoid any potential confounding effect due to differences in the difficulty to learn the novel word forms.

-As a filler task, the authors used a N-back working memory task, where participants had to indicate if a given letter was presented 2 trials before. I wonder why the authors didn’t use a non-linguistic task (e.g., a N-back task where participants had to indicate if a given picture was presented “n” trials before). A task involving letters may interfere with the following recall and recognition tasks, decreasing performance.

-The way in which the recall and recognition tasks were conducted was not the most appropriate. In the procedure used here, the recall of each word was followed by its recognition. In each trial of the recognition task, participants were presented with the correct word and with another word from the experimental setting (lure). Especially in the initial trials, this procedure involves an additional exposure to the words (the lures) before participants are asked to recall them in a subsequent recall trial. This may produce either facilitation or interference effects during recall. Such effects would be avoided if a two-steps procedure was used. Concretely, there should be a recall phase of all the items followed by a recognition phase of all the items. I would like to know why the authors didn’t choose this procedure, which is more common in this field of research.

-The numbers included in the design and stimuli section (page 14) are rather confusing. The authors state that the soundtracks of each of the two videos consisted of 28 sentences in Spanish containing a total of 12 targets. I don’t understand where these numbers come from. There are 12 targets, 4 of which are presented once, 4 of which are presented 4 times and 4 of which are presented 8 times. If I am not wrong, this makes a total number of 52 sentences.

-I would like the authors to include an appendix with the experimental sentences, or at least a table with some example sentences.

-In sum, the rationale for the above methodological choices has to be provided and the limitations above mentioned should be acknowledged in the paper.

Results:

Recall and recognition accuracy are presented as percentages of errors. I would prefer that variable to be named “percentage of errors”, or “error rate”, not “recall/recognition accuracy”.

Discussion:

-According to the authors, the results demonstrate that words containing language unspecified letter combinations are better learned than words that are orthographically marked. However, this is not a general pattern. This is true only for recall, but not for accuracy in word recognition. The authors also state that the learning of pseudowords increases linearly with repetitions, while this is not the case for nonwords. Again, this is only true for recall. Exposure produces an advantage in recognition for both pseudowords and nonwords. The summary of the results has to reflect these differences between recall and recognition.

-The authors have to discuss in depth those differences. In fact, the paper cited in the manuscript (Vander Beken and Brysbaert) reports an impairment in recall, but not in recognition (a true-false judgment test), for texts studied in L2 in comparison to L1. Apart from citing that paper, possible reasons for the different pattern of results between recall and recognition should be provided.

6. PLOS authors have the option to publish the peer review history of their article (what does this mean?). If published, this will include your full peer review and any attached files.

Reviewer #1: No

Reviewer #2: No

---

## [Author Response · Author response to Decision Letter 0]

9 Jan 2021

Dear Prof. Hinojosa,

First and foremost, we would like to thank you and the Reviewers for taking the time to assess our paper and for the interesting points they raised. The most significant modification made to the manuscript is the inclusion of a new experiment (Experiment 2 in the new version) along the lines suggested by the Reviewers, whose results fully replicate those obtained in the original experiment (now Experiment 1) and extend them to a more realistic incidental learning scenario. We’re very grateful to the Reviewers, since we sincerely believe that the inclusion of this experiment has significantly improved the quality and impact of the article. In the following pages, we have addressed each of the Reviewers’ concerns, one-by-one. All the changes have been marked in red in the new version of the manuscript.

Reviewer #1

Q: I like the paper and the idea is interesting, but the experiment suffers from several problems that need to be improved before publication. The easy path would be running a second better controlled experiment.

A: We are very grateful for the suggestion, and we have run a second experiment along the lines proposed by the Reviewer. In this second experiment, participants were exposed to videos in which aural and visual information was coherent. Also, novel words were embedded in the complete utterance displayed as on-screen text. By doing so, we tried to reproduce a more naturalistic incidental learning setting. 

Q: The procedure does not correspond to an incidental learning situation. Usually, exposures to linguistic stimuli are made through a single modality. In the case subtitled films, this is done by image, sound and text. In this case images (landscapes) were used, but they did not contain references to linguistic material. What then was the function of the images? It could have been done with text and audio only.

A: The reason why we included images of landscapes was to make/encourage participants to pay attention to the screen and thus be able to notice the subtitles. Nevertheless, in Experiment 2 we followed the Reviewer’s suggestions, modifying and including images of the targets, so participants were presented with the audio in their L1, the subtitles in the L1 except for the critical words that were shown in the ‘new language’, and the corresponding supporting visual information. 

Q: The authors showed the target words by text, instead of the entire utterance that would be the logical choice. In this way participants' attention is now drawn to these words, so that learning is no longer incidental.

A: We acknowledge this flaw and we addressed this issue in the second experiment, in which novel words were embedded in the entire utterance. We thank the Reviewer for this suggestion. 

Q: There are very few items per condition.

A: The number of items is similar to that of previous studies such as Hulme, Barsky and Rodd (2019), or Frances, Martin and Duñabeitia (2020). Nonetheless, we agree with the Reviewer that ideally one should test a larger number of items to grant generalization of the results, but it should be kept in mind that the direct recall was already low, so the inclusion of additional items would have led to a floor recall effect. In the current version of the manuscript the number of items per condition has been pointed out as a limitation of the study.

Q: The tasks used for the assessment only measure recall and recognition but not semantic and orthographic integration.

A: The Reviewer is completely right in pointing this out. In this study we have sticked to the procedure used in many earlier studies on incidental learning of vocabulary that have also tested recall and recognition. We understand that additional insights from different tasks measuring semantic integration would have been useful to draw stronger conclusions, but here we decided to replicate the procedure used by preceding studies from our group and from other groups testing incidental vocabulary and content learning. However, we’d like to mention that we’re currently trying to address this issue testing different tasks and paradigms in a new series of studies that are under preparation, with the aim of exploring semantic clustering and repetition effects on incidental vocabulary learning across tasks.

Q: There are no examples of the materials: Do the sentences form a coherent text or are they unrelated sentences?

A: All 52 sentences include objective information about the fruits and animals, respectively. They do not form a cohesive text, but they provide meaningful information about the items. We provide the materials in the Appendix of the article. 

Reviewer #2

Q: Ecological experimental paradigm. The authors state that their paradigm mimicks incidental vocabulary learning in naturalistic learning contexts. Concretely, they compare the paradigm used here with watching television or videos with subtitles. This similarity is evidenced by the title of the paper “Incidental vocabulary learning with subtitles in a new language”. I do not agree, however, with this comparison. It seems to me that this procedure does not mimic ecological incidental vocabulary learning. In my opinion, being exposed to an audiotape that has not any relationship with a series of images (landscapes) that are displayed on the screen, where a single word in an unknown language is presented, is not very ecological. This is rather an artificial situation, different from watching films with subtitles, where the auditory and the visual messages are integrated. Therefore, the autors should temper their claims about the ecological nature of the paradigm. I wonder why the authors didn’t use a more realistic situation, in which the content of the soundtrack is integrated with the content of the visual scenes. 

A: The Reviewer is completely right, and we highly appreciate these comments. In this revised version of the manuscript, we have added a new experiment (Experiment 2) in which most of the flaws that undermined the ecological validity of the first trial are addressed. In this second experiment included in this new version of the paper, we included images of the corresponding fruits and animals, instead of landscapes, integrated with the soundtrack. Thus, the auditory, textual and visual information matched. We sincerely hope that the results of this new experiment will convincingly speak for an incidental nature of the vocabulary learning, and we thank the Reviewer for the kind suggestions.

Q: I would like to know also why they didn’t present the written novel word (pseudoword or nonword) embedded in the Spanish sentence, as a way to facilitate the integration of its meaning.

A: In Experiment 2 we embedded the novel words within the entire utterances, mimicking a more realistic setting. Thanks for suggesting this change, which has helped us improving the study and the article.

Q: Participants. More information about the linguistic profile of the participants is needed. Considering that they are majoring in modern languages, they probably know several languages other than Spanish. Furthermore, they may have a high proficiency level in some of them. This information is relevant for two reasons. On the one hand, it has been demonstrated that bilinguals have a greater facility to acquire vocabulary in an unknown language than monolinguals (Kaushanskaya and Marian, 2009). This might explain the good performance observed in the recognition task (at least in the 4-8 exposures conditions). On the other hand, it is relevant to know if the letter combinations that are not allowed in Spanish (illegal bigrams) are legal combinations in any other of the languages spoken by the participants. If that is the case, this might have reduced the difference between pseudowords and nonwords for some participants.

A: We are very thankful for this comment, even though we respectfully disagree with the potential generalized impact of multilingualism in vocabulary learning. While the Reviewer is right in pointing out that some preceding studies have shown that bilinguals outperform monolinguals in word learning, it should be kept in mind that 1) there are very few studies showing such an advantage, 2) the studies focus on children samples, and 3) recent data have challenged some of those claims. A very recent article by Borragán, de Bruin, Havas, de Diego-Balaguer, Vulchanova, Vulchanov and Duñabeitia (in press in Applied Psycholinguistics) has shown that differences in word learning in children are not the consequence of bilingualism as such, but rather depend on the specific language combinations spoken by the bilinguals, making the hypotheses about the potential role of additional languages in the observed results of the current study complex and unclear. The present study was conducted with adult samples, and while they could admittedly know other languages than Spanish and acknowledging that it would be certainly interesting to analyse then effects of their competence in foreign languages on their recognition and recall abilities in incidental vocabulary learning, we could not assess their level of foreign language proficiency as part of the current study. Besides, while some of the participants were majoring in Modern Languages, the majority of the participants were majoring in other non-linguistic degrees, such as Psychology and Education. But more importantly, we asked for the languages in which the participants had a medium or high level according to their own perception, and all but two reported English as an additional language. Critically, none of the crucial bigrams of the nonwords exist in English (namely, MG, JL, ZF, GX, JR and WZ), thus minimizing any potential impact of the knowledge of this language. Anyhow, this comment will be considered in future studies, and we apologize for not being able to report additional data on the participants’ foreign language skills, but these were not part of the initial test battery approved by the Research Board. 

Q: Materials/Procedure. The number of items in each condition is very low. This is a 2x3 design, including a total amount of 12 words. This means that there are only 2 items per condition. I am aware that participants in this type of experiments cannot learn many new words, but this does not justify such a small number of items. In my opinion, this is the main limitation of the study. The authors might have used a more extensive training procedure, as other authors have done. Alternatively, they might have manipulated the frequency of exposure in a between-participants design. The results obtained here with such a small number of items may be restricted to this particular set of stimuli. The authors have to carry out a by-items analysis to examine if the effects can be generalized over items. This analysis has to be included in the paper, together with the by-participants analysis. All the participants were presented with the same 4 novel words in the 1, 4 and 8 exposures conditions. I wonder why the authors didn’t counterbalance the set of words included in these three conditions across participants. This would have been the most suitable approach to avoid any potential confounding effect due to differences in the difficulty to learn the novel word forms.

A: While some studies exploring vocabulary learning outside discourse context have used more test items, other studies in which the novel words have been embedded in larger texts have used a number of items similar to that used in this study, like Hulme, Barsky and Rodd (2019), or Frances, Martin and Duñabeitia (2020). Nonetheless, we agree with the Reviewer that ideally one should test a larger number of items to grant generalization of the results, but it should be kept in mind that the direct recall was already low, so the inclusion of additional items would have led to a floor recall effect. We fully understand that a by-item analysis could be of interest, but similar to the case found in earlier studies referenced in the manuscript testing incidental vocabulary learning, the current design does not allow for this approach, given that the number of items per specific cell does not meet the minimal requirements for such a statistical test. Anyhow, we have explicitly mentioned this as a potential limitation of the study in the General Discussion section of the manuscript

Q: As a filler task, the authors used a N-back working memory task, where participants had to indicate if a given letter was presented 2 trials before. I wonder why the authors didn’t use a non-linguistic task (e.g., a N-back task where participants had to indicate if a given picture was presented “n” trials before). A task involving letters may interfere with the following recall and recognition tasks, decreasing performance.

A: This comment has been taken into account in the second experiment, in which we used a non-linguistic task instead of the one used in the first experiment. We thank the Reviewer for this suggestion, and we refer to this point in the Procedure section of Experiment 2.

Q: The way in which the recall and recognition tasks were conducted was not the most appropriate. In the procedure used here, the recall of each word was followed by its recognition. 

In each trial of the recognition task, participants were presented with the correct word and with another word from the experimental setting (lure). Especially in the initial trials, this procedure involves an additional exposure to the words (the lures) before participants are asked to recall them in a subsequent recall trial. This may produce either facilitation or interference effects during recall. Such effects would be avoided if a two-steps procedure was used. Concretely, there should be a recall phase of all the items followed by a recognition phase of all the items. I would like to know why the authors didn’t choose this procedure, which is more common in this field of research.

A: We appreciate the Reviewer’s comment, and we took it into account when designing Experiment 2. We amended this in the second experiment, where we modified the order of the recall and the recognition tasks, so that participants had to recall all the targets first, and then complete a recognition task. This change has been also indicated in the Procedure section of Experiment 2.

Q: The numbers included in the design and stimuli section (page 14) are rather confusing. The authors state that the soundtracks of each of the two videos consisted of 28 sentences in Spanish containing a total of 12 targets. I don’t understand where these numbers come from. There are 12 targets, 4 of which are presented once, 4 of which are presented 4 times and 4 of which are presented 8 times. If I am not wrong, this makes a total number of 52 sentences.

A: We apologize for not having explained this correctly in the previous version of the manuscript, and we have tried to do so in the current one. The soundtrack of each of the two videos consisted of 28 sentences in Spanish, 26 of which provided information regarding the critical words. The remaining 2 sentences were included for opening (1) and closing (1) purposes. The 26 critical sentences of each video correspond to the 8 instances of the high-frequency nonword, the 8 instances of the high-frequency pseudoword, the 4 instances of the medium-frequency nonword, the 4 instances of the medium-frequency pseudoword, the single instance of the low-frequency nonword, and the single instance of the low-frequency pseudoword.

Q: I would like the authors to include an appendix with the experimental sentences, or at least a table with some example sentences.

A: In the current version of the article an appendix has been included.

Q: In sum, the rationale for the above methodological choices has to be provided and the limitations above mentioned should be acknowledged in the paper.

A: We have added a section discussing the limitations of the current study in the General Discussion section of the article.

Q: Results. Recall and recognition accuracy are presented as percentages of errors. I would prefer that variable to be named “percentage of errors”, or “error rate”, not “recall/recognition accuracy”.

A: We have changed the terminology to “percentage of errors” so that there is no confusion. 

Q: Discussion. According to the authors, the results demonstrate that words containing language unspecified letter combinations are better learned than words that are orthographically marked. However, this is not a general pattern. This is true only for recall, but not for accuracy in word recognition. The authors also state that the learning of pseudowords increases linearly with repetitions, while this is not the case for nonwords. Again, this is only true for recall. Exposure produces an advantage in recognition for both pseudowords and nonwords. The summary of the results has to reflect these differences between recall and recognition. The authors have to discuss in depth those differences. In fact, the paper cited in the manuscript (Vander Beken and Brysbaert) reports an impairment in recall, but not in recognition (a true-false judgment test), for texts studied in L2 in comparison to L1. Apart from citing that paper, possible reasons for the different pattern of results between recall and recognition should be provided.

A: The differences between recall and recognition with regard to the interaction between frequency and type of item are now addressed and interpreted in the current version of the manuscript. We are very thankful to the Reviewer for suggesting different potential explanations, and we have tried to elaborate along those lines in the General Discussion section of the manuscript.

---

## [Decision Letter · Decision Letter 1]

29 Jan 2021

Incidental vocabulary learning with subtitles in a new language: Orthographic markedness and number of exposures

PONE-D-20-13468R1

Dear Dr. Pérez Serrano,

The reviewers were satisfied with your responses to their previous concerns. Thus, we’re pleased to inform you that your manuscript has been judged scientifically suitable for publication and will be formally accepted for publication once it meets all outstanding technical requirements.

Kind regards,

José A Hinojosa, Ph.D.

Academic Editor

PLOS ONE

Additional Editor Comments (optional):

Reviewers' comments:

Reviewer's Responses to Questions

**Comments to the Author**

1. If the authors have adequately addressed your comments raised in a previous round of review and you feel that this manuscript is now acceptable for publication, you may indicate that here to bypass the “Comments to the Author” section, enter your conflict of interest statement in the “Confidential to Editor” section, and submit your "Accept" recommendation.

Reviewer #1: (No Response)

Reviewer #2: All comments have been addressed

2. Is the manuscript technically sound, and do the data support the conclusions?

Reviewer #1: (No Response)

Reviewer #2: (No Response)

3. Has the statistical analysis been performed appropriately and rigorously? 

Reviewer #1: (No Response)

Reviewer #2: (No Response)

4. Have the authors made all data underlying the findings in their manuscript fully available?

Reviewer #1: (No Response)

Reviewer #2: (No Response)

5. Is the manuscript presented in an intelligible fashion and written in standard English?

Reviewer #1: (No Response)

Reviewer #2: (No Response)

6. Review Comments to the Author

Reviewer #1: The authors did a very good job in this revised version, including a new

experiment, and I recommend publication.

Reviewer #2: (No Response)

7. PLOS authors have the option to publish the peer review history of their article (what does this mean?). If published, this will include your full peer review and any attached files.

Reviewer #1: **Yes: **Ana Marcet

Reviewer #2: No

---

## [Editor Report · Acceptance letter]

2 Feb 2021

PONE-D-20-13468R1 

Incidental vocabulary learning with subtitles in a new language: Orthographic markedness and number of exposures 

Dear Dr. Pérez Serrano:

I'm pleased to inform you that your manuscript has been deemed suitable for publication in PLOS ONE. Congratulations! Your manuscript is now with our production department. 

Kind regards, 

on behalf of

Dr. José A Hinojosa 

Academic Editor

PLOS ONE